# Learning Efficient Tensor Representations with Ring Structure Networks

**Qibin Zhao**
RIKEN Center for Advanced Intelligence Project
Tokyo, Japan
qibin.zhao@riken.jp

**Masashi Sugiyama**
RIKEN Center for Advanced Intelligence Project
& The University of Tokyo
Tokyo, Japan
sugi@k.u-tokyo.ac.jp

**Longhao Yuan**
RIKEN Center for Advanced Intelligence Project
& Saitama Institute of Technology, Japan
longhao.yuan@riken.jp

**Andrzej Cichocki**
SKOLTECH, Moscow, Russia
& RIKEN BSI, Japan
a.cichocki@skoltech.ru

## Abstract

*Tensor train (TT) decomposition* is a powerful representation for high-order tensors, which has been successfully applied to various machine learning tasks in recent years. In this paper, we propose a more generalized tensor decomposition with ring structure network by employing circular multilinear products over a sequence of lower-order core tensors, which is termed as TR representation. Several learning algorithms including blockwise ALS with adaptive tensor ranks and SGD with high scalability are presented. Furthermore, the mathematical properties are investigated, which enables us to perform basic algebra operations in a computationally efficiently way by using TR representations. Experimental results on synthetic signals and real-world datasets demonstrate the effectiveness of TR model and the learning algorithms. In particular, we show that the structure information and high-order correlations within a 2D image can be captured efficiently by employing an appropriate tensorization and TR decomposition.

## 1 Introduction

*Tensor decompositions* aim to represent a higher-order (or multi-dimensional) data as a multilinear product of several latent factors, which attracted considerable attentions in machine learning (Yu & Liu, 2016; Anandkumar et al., 2014; Romera-Paredes et al., 2013; Kanagawa et al., 2016; Yang et al., 2017) and signal processing (Zhou et al., 2016) in recent years. For a $d$th-order tensor with "square" core tensor of size $r$, standard tensor decompositions are the *canonical polyadic (CP) decomposition* (Goulart et al., 2015) which represents data as a sum of rank-one tensors by $\mathcal{O}(dnr)$ parameters and *Tucker decomposition* (De Lathauwer et al., 2000; Xu et al., 2012; Wu et al., 2014; Zhe et al., 2016) which represents data as a core tensor and several factor matrices by $\mathcal{O}(dnr + r^d)$ parameters. In general, CP decomposition provides a compact representation but with difficulties in finding the optimal solution, while Tucker decomposition is stable and flexible but its number of parameters scales exponentially to the tensor order.

Recently, *tensor networks* have emerged as a powerful tool for analyzing very high-order tensors (Cichocki et al., 2016). A powerful tensor network is *tensor train / matrix product states* (TT/MPS) representation (Oseledets, 2011), which requires $\mathcal{O}(dnr^2)$ parameters and avoid the curse of dimensionality through a particular geometry of low-order contracted tensors. TT representation has been applied to model weight parameters in deep neural network and nonlinear kernel learning (Novikov et al., 2015; Stoudenmire & Schwab, 2016; Tsai et al., 2016), achieving a significant compression factor and scalability. It also has been successfully used for feature learning and classification (Bengua et al., 2015). To fully explore the advantages of tensor algebra, the key step is to efficiently represent the real-world dataset by tensor networks, which is not well studied. In addition, there are some limitations of TT including that i) the constraint on TT-ranks, i.e., $r_1 = r_{d+1} = 1$, leads to the

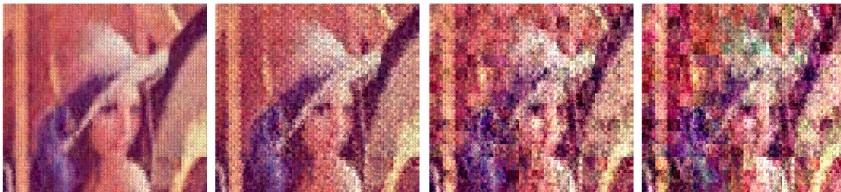

Figure 1: The effects of noise corrupted tensor cores. From left to right, each figure shows noise corruption by adding noise to one specific tensor core.

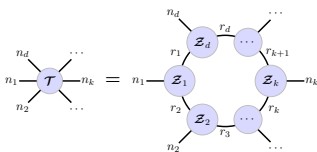

Figure 2: A graphical representation of tensor ring decomposition.

limited representation ability and flexibility; ii) TT-ranks are bounded by the rank of k-unfolding matricization, which might not be optimal; iii) the permutation of data tensor will yield an inconsistent solution, i.e., TT representations and TT-ranks are sensitive to the order of tensor dimensions. Hence, finding the optimal permutation remains a challenging problem.

In this paper, we introduce a new structure of tensor networks, which can be considered as a generalization of TT representations. First of all, we relax the condition over TT-ranks, i.e., $r_1 = r_{d+1} = 1$, leading to an enhanced representation ability. Secondly, the strict ordering of multilinear products between cores should be alleviated. Third, the cores should be treated equivalently by making the model symmetric. To this end, we add a new connection between the first and the last core tensors, yielding a circular tensor products of a set of cores (see Fig. 2). More specifically, we consider that each tensor element is approximated by performing a trace operation over the sequential multilinear products of cores. Since the trace operation ensures a scalar output, $r_1 = r_{d+1} = 1$ is not necessary. In addition, the cores can be circularly shifted and treated equivalently due to the properties of the trace operation. We call this model *tensor ring (TR) decomposition* and its cores *tensor ring (TR) representations*. To learn TR representations, we firstly develop a non-iterative TR-SVD algorithm that is similar to TT-SVD algorithm (Oseledets, 2011). To find the optimal lower TR-ranks, a block-wise ALS algorithms is presented. Finally, we also propose a scalable algorithm by using stochastic gradient descend, which can be applied to handling large-scale datasets.

Another interesting contribution is that we show the intrinsic structure or high order correlations within a 2D image can be captured more efficiently than SVD by converting 2D matrix to a higher order tensor. For example, given an image of size $I \times J$, we can apply an appropriate tensorization operation (see details in Sec. 5.2) to obtain a fourth order tensor, of which each mode controls one specific scale of resolution. To demonstrate this, Fig. 1 shows the effects caused by noise corruption of specific tensor cores. As we can see, the first mode corresponds to the small-scale patches, while the 4th-mode corresponds to the large-scale partitions. We have shown in Sec. 5.2 that TR model can represent the image more efficiently than the standard SVD.

## 2 TENSOR RING DECOMPOSITION

The TR decomposition aims to represent a high-order (or multi-dimensional) tensor by a sequence of 3rd-order tensors that are multiplied circularly. Specifically, let $\mathcal{T}$ be a $d$th-order tensor of size $n_1 \times n_2 \times \cdots \times n_d$, denoted by $\mathcal{T} \in \mathbb{R}^{n_1 \times \cdots \times n_d}$, TR representation is to decompose it into a sequence of latent tensors $\mathcal{Z}_k \in \mathbb{R}^{r_k \times n_k \times r_{k+1}}$, $k = 1, 2, \ldots, d$, which can be expressed in an element-wise form given by

$$T(i_1, i_2, \ldots, i_d) = \text{Tr}\left\{\mathbf{Z}_1(i_1)\mathbf{Z}_2(i_2)\cdots\mathbf{Z}_d(i_d)\right\} = \text{Tr}\left\{\prod_{k=1}^{d}\mathbf{Z}_k(i_k)\right\}. \tag{1}$$

$T(i_1, i_2, \ldots, i_d)$ denotes the $(i_1, i_2, \ldots, i_d)$th element of the tensor. $\mathbf{Z}_k(i_k)$ denotes the $i_k$th lateral slice matrix of the latent tensor $\boldsymbol{\mathcal{Z}}_k$, which is of size $r_k \times r_{k+1}$. Note that any two adjacent latent tensors, $\boldsymbol{\mathcal{Z}}_k$ and $\boldsymbol{\mathcal{Z}}_{k+1}$, have a common dimension $r_{k+1}$ on their corresponding modes. The last latent tensor $\boldsymbol{\mathcal{Z}}_d$ is of size $r_d \times n_d \times r_1$, i.e., $r_{d+1} = r_1$, which ensures the product of these matrices is a square matrix. These prerequisites play key roles in TR decomposition, resulting in some important numerical properties. For simplicity, the latent tensor $\boldsymbol{\mathcal{Z}}_k$ can also be called the $k$th-*core* (or *node*). The size of cores, $r_k, k = 1, 2, \ldots, d$, collected and denoted by a vector $\mathbf{r} = [r_1, r_2, \ldots, r_d]^T$, are called *TR-ranks*. From (1), we can observe that $T(i_1, i_2, \ldots, i_d)$ is equivalent to the trace of a sequential product of matrices $\{\mathbf{Z}_k(i_k)\}$. Based on (1), we can also express TR decomposition in the tensor form, given by

$$\boldsymbol{\mathcal{T}} = \sum_{\alpha_1, \ldots, \alpha_d=1}^{r_1, \ldots, r_d} \mathbf{z}_1(\alpha_1, \alpha_2) \circ \mathbf{z}_2(\alpha_2, \alpha_3) \circ \cdots \circ \mathbf{z}_d(\alpha_d, \alpha_1),$$

where the symbol '$\circ$' denotes the outer product of vectors and $\mathbf{z}_k(\alpha_k, \alpha_{k+1}) \in \mathbb{R}^{n_k}$ denotes the $(\alpha_k, \alpha_{k+1})$th mode-2 fiber of tensor $\boldsymbol{\mathcal{Z}}_k$. The number of parameters in TR representation is $\mathcal{O}(dnr^2)$, which is linear to the tensor order $d$ as in TT representation.

The TR representation can also be illustrated graphically by a linear tensor network as shown in Fig. 2. A node represents a tensor (including a matrix and a vector) whose order is denoted by the number of edges. The number by an edge specifies the size of each mode (or dimension). The connection between two nodes denotes a multilinear product operator between two tensors on a specific mode. This is also called *tensor contraction*, which corresponds to the summation over the indices of that mode. It should be noted that $\boldsymbol{\mathcal{Z}}_d$ is connected to $\boldsymbol{\mathcal{Z}}_1$ by the summation over the index $\alpha_1$, which is equivalent to the trace operation. For simplicity, we denote TR decomposition by $\boldsymbol{\mathcal{T}} = \Re(\boldsymbol{\mathcal{Z}}_1, \boldsymbol{\mathcal{Z}}_2, \ldots, \boldsymbol{\mathcal{Z}}_d)$.

**Theorem 1** (Circular dimensional permutation invariance). *Let $\boldsymbol{\mathcal{T}} \in \mathbb{R}^{n_1 \times n_2 \times \cdots \times n_d}$ be a dth-order tensor and its TR decomposition is given by $\boldsymbol{\mathcal{T}} = \Re(\boldsymbol{\mathcal{Z}}_1, \boldsymbol{\mathcal{Z}}_2, \ldots, \boldsymbol{\mathcal{Z}}_d)$. If we define $\overleftarrow{\boldsymbol{\mathcal{T}}}^k \in \mathbb{R}^{n_{k+1} \times \cdots \times n_d \times n_1 \times \cdots \times n_k}$ as the circularly shifted version along the dimensions of $\boldsymbol{\mathcal{T}}$ by k, then we have $\overleftarrow{\boldsymbol{\mathcal{T}}}^k = \Re(\boldsymbol{\mathcal{Z}}_{k+1}, \ldots, \boldsymbol{\mathcal{Z}}_d, \boldsymbol{\mathcal{Z}}_1, \ldots \boldsymbol{\mathcal{Z}}_k)$.*

A proof of Theorem 1 is provided in Appendix B.1.

It should be noted that circular dimensional permutation invariance is an essential feature that distinguishes TR decomposition from TT decomposition. For TT decomposition, the product of matrices must keep a strictly sequential order, yielding that the tensor with a circular dimension shifting does not correspond to the shifting of tensor cores.

## 3 LEARNING ALGORITHMS

### 3.1 SEQUENTIAL SVDS

We propose the first algorithm for computing the TR decomposition using $d$ sequential SVDs. This algorithm will be called the *TR-SVD algorithm*.

**Theorem 2.** *Let us assume $\boldsymbol{\mathcal{T}}$ can be represented by a TR decomposition. If the $k$-unfolding matrix $\mathbf{T}_{\langle k \rangle}$ has $Rank(\mathbf{T}_{\langle k \rangle}) = R_{k+1}$, then there exists a TR decomposition with TR-ranks $\mathbf{r}$ which satisfies that $\exists k, r_1 r_{k+1} \leq R_{k+1}$.*

*Proof.* We can express TR decomposition in the form of $k$-unfolding matrix,

$$T_{\langle k \rangle}(\overline{i_1 \cdots i_k}, \overline{i_{k+1} \cdots i_d}) = \mathrm{Tr}\left\{\prod_{j=1}^{k} \mathbf{Z}_j(i_j) \prod_{j=k+1}^{d} \mathbf{Z}_j(i_j)\right\} = \left\langle \mathrm{vec}\left(\prod_{j=1}^{k} \mathbf{Z}_j(i_j)\right), \mathrm{vec}\left(\prod_{j=d}^{k+1} \mathbf{Z}_j^T(i_j)\right)\right\rangle.$$
$$(2)$$

It can also be rewritten as

$$T_{\langle k \rangle}(\overline{i_1 \cdots i_k}, \overline{i_{k+1} \cdots i_d}) = \sum_{\alpha_1 \alpha_{k+1}} Z^{\leq k}\left(\overline{i_1 \cdots i_k}, \overline{\alpha_1 \alpha_{k+1}}\right) Z^{>k}\left(\overline{\alpha_1 \alpha_{k+1}}, \overline{i_{k+1} \cdots i_d}\right), \quad (3)$$

where we defined the subchain by merging multiple linked cores as $\mathbf{Z}^{<k}(\overline{i_1 \cdots i_{k-1}}) = \prod_{j=1}^{k-1} \mathbf{Z}_j(i_j)$ and $\mathbf{Z}^{>k}(\overline{i_{k+1} \cdots i_d}) = \prod_{j=k+1}^{d} \mathbf{Z}_j(i_j)$. Hence, we can obtain $\mathbf{T}_{\langle k \rangle} = \mathbf{Z}_{(2)}^{\leq k}(\mathbf{Z}_{[2]}^{>k})^T$, where the subchain $\mathbf{Z}_{(2)}^{\leq k}$ is of size $\prod_{j=1}^{k} n_j \times r_1 r_{k+1}$, and $\mathbf{Z}_{[2]}^{>k}$ is of size $\prod_{j=k+1}^{d} n_j \times r_1 r_{k+1}$. Since the rank of $\mathbf{T}_{\langle k \rangle}$ is $R_{k+1}$, we can obtain $r_1 r_{k+1} \leq R_{k+1}$. $\qquad\square$

According to (2) and (3), TR decomposition can be written as

$$T_{\langle 1 \rangle}(i_1, \overline{i_2 \cdots i_d}) = \sum_{\alpha_1, \alpha_2} Z^{\leq 1}(i_1, \overline{\alpha_1 \alpha_2}) Z^{>1}(\overline{\alpha_1 \alpha_2}, \overline{i_2 \cdots i_d}).$$

Since the low-rank approximation of $\mathbf{T}_{\langle 1 \rangle}$ can be obtained by the truncated SVD, which is $\mathbf{T}_{\langle 1 \rangle} = \mathbf{U}\mathbf{\Sigma}\mathbf{V}^T + \mathbf{E}_1$, the first core $\mathbf{\mathcal{Z}}_1 (i.e., \mathbf{\mathcal{Z}}^{\leq 1})$ of size $r_1 \times n_1 \times r_2$ can be obtained by the proper reshaping and permutation of $\mathbf{U}$ and the subchain $\mathbf{\mathcal{Z}}^{>1}$ of size $r_2 \times \prod_{j=2}^{d} n_j \times r_1$ is obtained by the proper reshaping and permutation of $\mathbf{\Sigma}\mathbf{V}^T$, which corresponds to the remaining $d-1$ dimensions of $\mathcal{T}$. Note that this algorithm use the similar strategy with TT-SVD (Oseledets, 2011), but the reshaping and permutations are totally different between them. Subsequently, we can further reshape the subchain $\mathbf{\mathcal{Z}}^{>1}$ as a matrix $\mathbf{Z}^{>1} \in \mathbb{R}^{r_2 n_2 \times \prod_{j=3}^{d} n_j r_1}$ which thus can be written as

$$Z^{>1}(\overline{\alpha_2 i_2}, \overline{i_3 \cdots i_d \alpha_1}) = \sum_{\alpha_3} Z_2(\overline{\alpha_2 i_2}, \alpha_3) Z^{>2}(\alpha_3, \overline{i_3 \cdots i_d \alpha_1}).$$

By applying truncated SVD, i.e., $\mathbf{Z}^{>1} = \mathbf{U}\mathbf{\Sigma}\mathbf{V}^T + \mathbf{E}_2$, we can obtain the second core $\mathbf{\mathcal{Z}}_2$ of size $(r_2 \times n_2 \times r_3)$ by appropriately reshaping $\mathbf{U}$ and the subchain $\mathbf{\mathcal{Z}}^{>2}$ by proper reshaping of $\mathbf{\Sigma}\mathbf{V}^T$. This procedure can be performed sequentially to obtain all $d$ cores $\mathbf{\mathcal{Z}}_k, k = 1, \ldots, d$.

As proved in (Oseledets, 2011), the approximation error by using such sequential SVDs is given by

$$\|\mathcal{T} - \Re(\mathbf{\mathcal{Z}}_1, \mathbf{\mathcal{Z}}_2, \ldots, \mathbf{\mathcal{Z}}_d)\|_F \leq \sqrt{\sum_{k=1}^{d-1} \|\mathbf{E}_k\|_F^2}.$$

Hence, given a prescribed relative error $\epsilon_p$, the truncation threshold $\delta$ can be set to $\frac{\epsilon_p}{\sqrt{d-1}}\|\mathcal{T}\|_F$. However, considering that $\|\mathbf{E}_1\|_F$ corresponds to two ranks including both $r_1$ and $r_2$, while $\|\mathbf{E}_k\|_F, \forall k > 1$ correspond to only one rank $r_{k+1}$. Therefore, we modify the truncation threshold as

$$\delta_k = \begin{cases} \sqrt{2}\epsilon_p \|\mathcal{T}\|_F / \sqrt{d} & k = 1, \\ \epsilon_p \|\mathcal{T}\|_F / \sqrt{d} & k > 1. \end{cases} \tag{4}$$

A pseudocode of the TR-SVD algorithm is summarized in Alg. 1. Note that the cores obtained by the TR-SVD algorithm are left-orthogonal, which is $\mathbf{Z}_{k\langle 2 \rangle}^T \mathbf{Z}_{k\langle 2 \rangle} = \mathbf{I}$ for $k = 2, \ldots, d-1$.

## 3.2 BLOCK-WISE ALTERNATING LEAST-SQUARES (ALS)

The ALS algorithm has been widely applied to various tensor decomposition models such as CP and Tucker decompositions (Kolda & Bader, 2009; Holtz et al., 2012). The main concept of ALS is optimizing one core while the other cores are fixed, and this procedure will be repeated until some convergence criterion is satisfied. Given a $d$th-order tensor $\mathcal{T}$, our goal is optimize the error function as

$$\min_{\mathbf{\mathcal{Z}}_1, \ldots, \mathbf{\mathcal{Z}}_d} \|\mathcal{T} - \Re(\mathbf{\mathcal{Z}}_1, \ldots, \mathbf{\mathcal{Z}}_d)\|_F. \tag{5}$$

According to the TR definition in (1), we have

$$T(i_1, i_2, \ldots, i_d) = \sum_{\alpha_1, \ldots, \alpha_d} Z_1(\alpha_1, i_1, \alpha_2) Z_2(\alpha_2, i_2, \alpha_3) \cdots Z_d(\alpha_d, i_d, \alpha_1)$$

$$= \sum_{\alpha_k, \alpha_{k+1}} \left\{ Z_k(\alpha_k, i_k, \alpha_{k+1}) Z^{\neq k}(\alpha_{k+1}, \overline{i_{k+1} \cdots i_d i_1 \cdots i_{k-1}}, \alpha_k) \right\},$$

where $\mathbf{Z}^{\neq k}(\overline{i_{k+1} \cdots i_d i_1 \ldots i_{k-1}}) = \prod_{j=k+1}^{d} \mathbf{Z}_j(i_j) \prod_{j=1}^{k-1} \mathbf{Z}_j(i_j)$ denotes a slice matrix of subchain tensor by merging all cores except $k$th core $\boldsymbol{\mathcal{Z}}_k$. Hence, the mode-$k$ unfolding matrix of $\boldsymbol{\mathcal{T}}$ can be expressed by

$$T_{[k]}(i_k, \overline{i_{k+1} \cdots i_d i_1 \cdots i_{k-1}}) = \sum_{\alpha_k \alpha_{k+1}} \left\{ Z_k(i_k, \overline{\alpha_k \alpha_{k+1}}) Z^{\neq k}(\overline{\alpha_k \alpha_{k+1}}, \overline{i_{k+1} \cdots i_d i_1 \cdots i_{k-1}}) \right\}.$$

By applying different mode-$k$ unfolding operations, we can obtain that $\mathbf{T}_{[k]} = \mathbf{Z}_{k(2)} \left( \mathbf{Z}_{[2]}^{\neq k} \right)^T$, where $\boldsymbol{\mathcal{Z}}^{\neq k}$ is a subchain obtained by merging $d-1$ cores.

The objective function in (5) can be optimized by solving $d$ subproblems alternatively. More specifically, having fixed all but one core, the problem reduces to a linear least squares problem, which is

$$\min_{\mathbf{Z}_{k(2)}} \left\| \mathbf{T}_{[k]} - \mathbf{Z}_{k(2)} \left( \mathbf{Z}_{[2]}^{\neq k} \right)^T \right\|_F, \quad k = 1, \ldots, d.$$

This optimization procedure must be repeated till the convergence, which is called TT-ALS.

Here, we propose a computationally efficient block-wise ALS (BALS) algorithm by utilizing truncated SVD, which facilitates the self-adaptation of ranks. The main idea is to perform the blockwise optimization followed by the separation of a block into individual cores. To achieve this, we consider merging two linked cores, e.g., $\boldsymbol{\mathcal{Z}}_k, \boldsymbol{\mathcal{Z}}_{k+1}$, into a block (or subchain) $\boldsymbol{\mathcal{Z}}^{(k,k+1)} \in \mathbb{R}^{r_k \times n_k n_{k+1} \times r_{k+2}}$. Thus, the subchain $\boldsymbol{\mathcal{Z}}^{(k,k+1)}$ can be optimized while leaving all cores except $\boldsymbol{\mathcal{Z}}_k, \boldsymbol{\mathcal{Z}}_{k+1}$ fixed. Subsequently, the subchain $\boldsymbol{\mathcal{Z}}^{(k,k+1)}$ can be reshaped into $\tilde{\mathbf{Z}}^{(k,k+1)} \in \mathbb{R}^{r_k n_k \times n_{k+1} r_{k+2}}$ and separated into a left-orthonormal core $\boldsymbol{\mathcal{Z}}_k$ and $\boldsymbol{\mathcal{Z}}_{k+1}$ by a truncated SVD:

$$\tilde{\mathbf{Z}}^{(k,k+1)} = \mathbf{U} \boldsymbol{\Sigma} \mathbf{V}^T = \mathbf{Z}_{k\langle 2 \rangle} \mathbf{Z}_{k+1\langle 1 \rangle}, \tag{6}$$

where $\mathbf{Z}_{k\langle 2 \rangle} \in \mathbb{R}^{r_k n_k \times r_{k+1}}$ is the 2-unfolding matrix of core $\boldsymbol{\mathcal{Z}}_k$, which can be set to $\mathbf{U}$, while $\mathbf{Z}_{k+1\langle 1 \rangle} \in \mathbb{R}^{r_{k+1} \times n_{k+1} r_{k+2}}$ is the 1-unfolding matrix of core $\boldsymbol{\mathcal{Z}}_{k+1}$, which can be set to $\boldsymbol{\Sigma} \mathbf{V}^T$. This procedure thus moves on to optimize the next block cores $\boldsymbol{\mathcal{Z}}^{(k+1,k+2)}, \ldots, \boldsymbol{\mathcal{Z}}^{(d-1,d)}, \boldsymbol{\mathcal{Z}}^{(d,1)}$ successively in the similar way. Note that since the TR model is circular, the $d$th core can also be merged with the first core yielding the block core $\boldsymbol{\mathcal{Z}}^{(d,1)}$.

The key advantage of our BALS algorithm is the rank adaptation ability which can be achieved simply by separating the block core into two cores via truncated SVD, as shown in (6). The truncated rank $r_{k+1}$ can be chosen such that the approximation error is below a certain threshold. One possible choice is to use the same threshold as in the TR-SVD algorithm, i.e., $\delta_k$ described in (4). However, the empirical experience shows that this threshold often leads to overfitting and the truncated rank is higher than the optimal rank. This is because the updated block $\boldsymbol{\mathcal{Z}}^{(k,k+1)}$ during ALS iterations is not a closed form solution and many iterations are necessary for convergence. To relieve this problem, we choose the truncation threshold based on both the current and the desired approximation errors, which is

$$\delta = \max \left\{ \epsilon \|\boldsymbol{\mathcal{T}}\|_F / \sqrt{d}, \ \epsilon_p \|\boldsymbol{\mathcal{T}}\|_F / \sqrt{d} \right\}.$$

A pseudo code of the BALS algorithm is described in Alg. 2.

## 3.3 STOCHASTIC GRADIENT DESCENT

For large-scale dataset, the ALS algorithm is not scalable due to the cubic time complexity in the target rank, while Stochastic Gradient Descent (SGD) shows high efficiency and scalability for matrix/tensor factorization (Gemulla et al., 2011; Maehara et al., 2016; Wang & Anandkumar, 2016). In this section, we present a scalable and efficient TR decomposition by using SGD, which is also suitable for online learning and tensor completion problems. To this end, we first provide the element-wise loss function, which is

$$L(\boldsymbol{\mathcal{Z}}_1, \boldsymbol{\mathcal{Z}}_2, \ldots, \boldsymbol{\mathcal{Z}}_d) = \frac{1}{2} \sum_{i_1, \ldots, i_d} \left\{ T(i_1, i_2, \ldots, i_d) - \mathrm{Tr} \left( \prod_{k=1}^{d} \mathbf{Z}_k(i_k) \right) \right\}^2 + \frac{1}{2} \lambda_k \|\mathbf{Z}_k(i_k)\|^2, \tag{7}$$

where $\lambda_k$ is the regularization parameters. The core idea of SGD is to randomly select one sample $\mathcal{T}(i_1, i_2, \ldots, i_d)$, then update the corresponding slice matrices $\mathbf{Z}_k(i_k), k = 1, \ldots, d$ from each latent core tensor $\mathcal{Z}_k$ based on the noisy gradient estimates by scaling up just one of local gradients, i.e. $\forall k = 1, \ldots, d$,

$$\frac{\partial L}{\partial \mathbf{Z}_k(i_k)} = - \left\{ T(i_1, i_2, \ldots, i_d) - \mathrm{Tr}\left( \prod_{k=1}^{d} \mathbf{Z}_k(i_k) \right) \right\} \left( \prod_{j=1, j \neq k}^{d} \mathbf{Z}_j(i_j) \right)^T + \lambda_k \mathbf{Z}_k(i_k), \quad (8)$$

We employ Adaptive Moment Estimation (Adam) method to compute adaptive learning rates for each parameter. Thus, the update rule for each core tensor is given by

$$\mathbf{Z}_k(i_k)^t = \mathbf{Z}_k^{t-1}(i_k) - \frac{\eta}{\sqrt{\mathbf{V}_t} + \epsilon} \mathbf{M}_t - \lambda_k \mathbf{Z}_k^{t-1}(i_k), \quad \forall k = 1, \ldots, d, \quad (9)$$

where $\mathbf{M}_t = \beta_1 \mathbf{M}_{t-1} + (1 - \beta_1) \frac{\partial L}{\partial \mathbf{Z}_k^t(i_k)}$ denotes an exponentially decaying average of past gradients and $\mathbf{V}_t = \beta_2 \mathbf{V}_{t-1} + (1 - \beta_2)(\frac{\partial L}{\partial \mathbf{Z}_k^t(i_k)})^2$ denotes exponentially decaying average of second moment of the gradients.

The SGD algorithm can be naturally applied to tensor completion problem, when the data points are sampled only from a sparse tensor. Furthermore, this also naturally gives an online TR decomposition. The batched versions, in which multiple local losses are averaged, are also feasible but often have inferior performance in practice. For each element $T(i_1, i_2, \ldots, i_d)$, the computational complexity of SGD is $\mathcal{O}(d^2 r^3)$. If we define $N = \prod_{k=1}^{d} n_k$ consecutive updates as one epoch of SGD, the computational complexity per SGD epoch is thus $\mathcal{O}(N d^2 r^3)$, which linearly scales to data size. As compared to ALS, which needs $\mathcal{O}(N d r^4 + d r^6)$, it is more efficient in terms of computational complexity for one epoch. The memory cost of TR-SVD, ALS, and SGD are $\mathcal{O}(n d r^2 + \prod_k n_k)$, $\mathcal{O}(n d r^2 + r^2 \prod_k n_k + r^4)$, and $\mathcal{O}(n d r^2 + r^2)$, respectively. Thus, TR-SGD requires much less memory. The convergence condition of SGD algorithm follows other stochastic tensor decompositions (Ge et al., 2015; Maehara et al., 2016).

In practice, how to choose the TR algorithms depends on the task and data at hand. TR-SVD is a non-iterative algorithm leading to the fast computation. The expected approximation error is required from user to choose TR-ranks automatically. However, the learned ranks is not necessary to be the optimal in terms of compression rate. TR-ALS is an iterative algorithm in which the fixed TR-ranks are required to be given by user. The computation time is highly related to TR-ranks, which is very efficient only for small TR-ranks. TR-BALS is an iterative algorithm that can choose TR-ranks in an adaptive manner given the expected approximation error. TR-SGD is an iterative algorithm that is suitable for a large-scale tensor with given TR-ranks. In particular, the memory cost of TR-SGD depends on only the size of core tensors rather than the size of data tensor.

## 4 PROPERTIES OF TR REPRESENTATION

By assuming that tensor data have been already represented as TR decompositions, i.e., a sequence of third-order cores, we justify and demonstrate that the basic operations on tensors, such as the *addition*, *multilinear product*, *Hadamard product*, *inner product* and *Frobenius norm*, can be performed efficiently by the appropriate operations on each individual cores. We have the following theorems:

**Property 1.** *Let $\mathcal{T}_1$ and $\mathcal{T}_2$ be dth-order tensors of size $n_1 \times \cdots \times n_d$. If TR decompositions of these two tensors are $\mathcal{T}_1 = \Re(\mathcal{Z}_1, \ldots, \mathcal{Z}_d)$ where $\mathcal{Z}_k \in \mathbb{R}^{r_k \times n_k \times r_{k+1}}$ and $\mathcal{T}_2 = \Re(\mathcal{Y}_1, \ldots, \mathcal{Y}_d)$ where $\mathcal{Y}_k \in \mathbb{R}^{s_k \times n_k \times s_{k+1}}$, then the addition of these two tensors, $\mathcal{T}_3 = \mathcal{T}_1 + \mathcal{T}_2$, can also be represented in the TR format given by $\mathcal{T}_3 = \Re(\mathcal{X}_1, \ldots, \mathcal{X}_d)$, where $\mathcal{X}_k \in \mathbb{R}^{q_k \times n_k \times q_{k+1}}$ and $q_k = r_k + s_k$. Each core $\mathcal{X}_k$ can be computed by*

$$\mathbf{X}_k(i_k) = \begin{pmatrix} \mathbf{Z}_k(i_k) & 0 \\ 0 & \mathbf{Y}_k(i_k) \end{pmatrix}, \quad \begin{matrix} i_k = 1, \ldots, n_k, \\ k = 1, \ldots, d. \end{matrix} \quad (10)$$

A proof of Property 1 is provided in Appendix B.2. Note that the sizes of new cores are increased and not optimal in general. This problem can be solved by the rounding procedure (Oseledets, 2011).

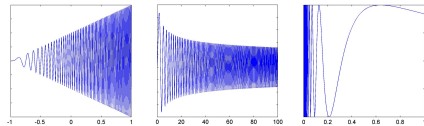

Figure 3: Highly oscillated functions. The left panel is $f_1(x) = (x+1)\sin(100(x+1)^2)$. The middle panel is Airy function: $f_2(x) = x^{-\frac{1}{4}}\sin(\frac{2}{3}x^{\frac{3}{2}})$. The right panel is Chirp function $f_3(x) = \sin\frac{x}{4}\cos(x^2)$.

**Property 2.** *Let $\mathcal{T} \in \mathbb{R}^{n_1 \times \cdots \times n_d}$ be a dth-order tensor whose TR representation is $\mathcal{T} = \Re(\mathcal{Z}_1, \ldots, \mathcal{Z}_d)$ and $\mathbf{u}_k \in \mathbb{R}^{n_k}, k = 1, \ldots, d$ be a set of vectors, then the multilinear products, denoted by $c = \mathcal{T} \times_1 \mathbf{u}_1^T \times_2 \cdots \times_d \mathbf{u}_d^T$, can be computed by the multilinear product on each cores, which is*

$$c = \Re(\mathbf{X}_1, \ldots, \mathbf{X}_d) \ where \ \mathbf{X}_k = \sum_{i_k=1}^{n_k} \mathbf{Z}_k(i_k)u_k(i_k). \tag{11}$$

A proof of Property 2 is provided in Appendix B.3. It should be noted that the computational complexity in the original tensor form is $\mathcal{O}(dn^d)$, while it reduces to $\mathcal{O}(dnr^2 + dr^3)$ that is linear to tensor order $d$ by using TR representation.

**Property 3.** *Let $\mathcal{T}_1$ and $\mathcal{T}_2$ be dth-order tensors of size $n_1 \times \cdots \times n_d$. If the TR decompositions of these two tensors are $\mathcal{T}_1 = \Re(\mathcal{Z}_1, \ldots, \mathcal{Z}_d)$ where $\mathcal{Z}_k \in \mathbb{R}^{r_k \times n_k \times r_{k+1}}$ and $\mathcal{T}_2 = \Re(\mathcal{Y}_1, \ldots, \mathcal{Y}_d)$ where $\mathcal{Y}_k \in \mathbb{R}^{s_k \times n_k \times s_{k+1}}$, then the Hadamard product of these two tensors, $\mathcal{T}_3 = \mathcal{T}_1 \circledast \mathcal{T}_2$, can also be represented in the TR format given by $\mathcal{T}_3 = \Re(\mathcal{X}_1, \ldots, \mathcal{X}_d)$, where $\mathcal{X}_k \in \mathbb{R}^{q_k \times n_k \times q_{k+1}}$ and $q_k = r_k s_k$. Each core $\mathcal{X}_k$ can be computed by*

$$\mathbf{X}_k(i_k) = \mathbf{Z}_k(i_k) \otimes \mathbf{Y}_k(i_k), \quad k = 1, \ldots, d. \tag{12}$$

*The inner product of two tensors can be computed by TR format $\langle \mathcal{T}_1, \mathcal{T}_2 \rangle = \Re(\mathbf{V}_1, \ldots, \mathbf{V}_d)$ where $\mathbf{V}_k = \sum_{i_k=1}^{n_k} \mathbf{Z}_k(i_k) \otimes \mathbf{Y}_k(i_k)$. Therefore, the Frobenius norm can be also computed by TR format using $\|\mathcal{T}\|_F = \sqrt{\langle \mathcal{T}, \mathcal{T} \rangle}$.*

A proof of Property 3 is provided in Appendix B.4. In contrast to $\mathcal{O}(n^d)$ in the original tensor form, the computational complexity is equal to $\mathcal{O}(dnq^2 + dq^3)$ that is linear to $d$ by using TR representation.

## 5 EXPERIMENTAL RESULTS

### 5.1 NUMERICAL ILLUSTRATION

We consider highly oscillating functions that can be approximated perfectly by a low-rank TT format (Khoromskij, 2015), as shown in Fig. 3. We firstly tensorize the functional vector resulting in a $d$th-order tensor of size $n_1 \times n_2 \times \cdots \times n_d$, where isometric size is usually preferred, i.e., $n_1 = n_2 = \cdots = n_d = n$, with the total number of elements denoted by $N = n^d$. The error bound (tolerance), denoted by $\epsilon_p = 10^{-3}$, is given as the stopping criterion for all compared algorithms. As shown in Table 1, TR-SVD and TR-BALS can obtain comparable results with TT-SVD in terms of compression ability. However, when noise is involved, TR model significantly outperforms TT model, indicating its more robustness to noises.

Table 1: The functional data $f_1(x), f_2(x), f_3(x)$ is tensorized to 10th-order tensor $(4 \times 4 \times \ldots \times 4)$. In the table, $\epsilon, \bar{r}, N_p$ denote relative error, average rank, and the total number of parameters, respectively.

| | $f_1(x)$ | | | | $f_2(x)$ | | | | $f_3(x)$ | | | | $f_1(x) + \mathcal{N}(0, \sigma), SNR = 60dB$ | | | |
|---|---|---|---|---|---|---|---|---|---|---|---|---|---|---|---|---|
| | $\epsilon$ | $\bar{r}$ | $N_p$ | Time (s) | $\epsilon$ | $\bar{r}$ | $N_p$ | Time (s) | $\epsilon$ | $\bar{r}$ | $N_p$ | Time (s) | $\epsilon$ | $\bar{r}$ | $N_p$ | Time (s) |
| TT-SVD | 3e-4 | 4.4 | 1032 | 0.17 | 3e-4 | 5 | 1360 | 0.16 | 3e-4 | 3.7 | 680 | 0.16 | 1e-3 | 16.6 | 13064 | 0.5 |
| TR-SVD | 3e-4 | 4.4 | 1032 | 0.17 | 3e-4 | 5 | 1360 | 0.28 | 5e-4 | 3.6 | 668 | 0.15 | 1e-3 | 9.7 | 4644 | 0.4 |
| TR-ALS | 3e-4 | 4.4 | 1032 | 13.2 | 3e-4 | 5 | 1360 | 18.6 | 8e-4 | 3.6 | 668 | 4.0 | 1e-3 | 4.4 | 1032 | 11.8 |
| TR-BALS | 9e-4 | 4.3 | 1052 | 4.6 | 8e-4 | 4.9 | 1324 | 5.7 | 5e-4 | 3.7 | 728 | 3.4 | 1e-3 | 4.2 | 1000 | 6.1 |

Table 2: The results under different shifts of dimensions on functional data $f_2(x)$ with error bound set at $10^{-3}$. For the 10th-order tensor, all 9 dimension shifts were considered and the average rank $\bar{r}$ is compared.

| | | | | | $\bar{r}$ | | | | |
|---|---|---|---|---|---|---|---|---|---|
| | 1 | 2 | 3 | 4 | 5 | 6 | 7 | 8 | 9 |
| TT | 5.2 | 5.8 | 6 | 6.2 | 7 | 7 | 8.5 | 14.6 | 8.4 |
| TR | 5 | 4.9 | 5 | 4.9 | 4.9 | 5 | 5 | 4.8 | 4.9 |

Figure 4: TR-SGD decomposition with TR-ranks of 12 on the 8th-order tensorization of an image. Iter: 50% indicates that only 50% elements are sampled for learning its TR representation. RSE indicates root relative square error $\|\hat{\mathcal{Y}} - \mathcal{Y}\|_F / \|\mathcal{Y}\|_F$.

It should be noted that TT representation has the property that $r_1 = r_{d+1} = 1$ and $r_k, k = 2, \ldots, d-1$ are bounded by the rank of $k$-unfolding matrix of $\mathbf{T}_{\langle k \rangle}$, which limits its generalization ability and consistency when the tensor modes have been shifted or permuted. To demonstrate this, we consider shifting the dimensions of $\mathcal{T}$ of size $n_1 \times \cdots \times n_d$ by $k$ times leading to $\overleftarrow{\mathcal{T}}^k$ of size $n_{k+1} \times \cdots \times n_d \times n_1 \times \cdots \times n_k$. As shown in Table 2, the average TT-ranks are varied dramatically along with the different shifts. In particular, when $k = 8$, $\bar{r}_{tt}$ becomes 14.6, resulting in a large number of parameters $N_p = 10376$. In contrast to TT, TR can obtain consistent and compact representation with $N_p = 1360$.

We also tested TR-ALS and TR-SGD algorithms on datasets which are generated by a TR model, in which the core tensors are randomly drawn from $\mathcal{N}(0, 1)$. As shown in Table 3, TR-SGD can achieve similar performance as TR-ALS in all cases. In particular, when data is relatively large-scale ($10^8$), TR-SGD can achieve relative error $\epsilon = 0.01$ by using 1% of data points only once.

Table 3: Results on synthetic data with fixed ranks $r_1 = r_2 = \cdots = 2$.

| Tensor size | TR-ALS | TR-SGD |
|---|---|---|
| $n = 10, d = 4$ | ($\epsilon = 0.01$, Epoch = 19) | ($\epsilon = 0.01$, Epoch = 10 ) |
| $n = 10, d = 6$ | ($\epsilon = 0.01$, Epoch = 10) | ($\epsilon = 0.01$, Epoch = 0.4 ) |
| $n = 10, d = 8$ | ($\epsilon = 0.05$, Epoch = 9) | ($\epsilon = 0.01$, Epoch = 0.01 ) |

## 5.2 Image Representation by Higher-order Tensor Decompositions

An image is naturally represented by a 2D matrix, on which SVD can provide the best low-rank approximation. However, the intrinsic structure and high-order correlations within the image is not well exploited by SVD. In this section, we show the tensorization of an image, yielding a higher-order tensor, and TR decomposition enable us to represent the image more efficiently than SVD. Given an image (e.g. 'Peppers') denoted by $\mathcal{Y}$ of size $I \times J$, we can reshape it as $I_1 \times I_2 \times \ldots \times I_d \times J_1 \times J_2 \times \ldots \times J_d$ followed by an appropriate permutation to $I_1 \times J_1 \times I_2 \times J_2 \ldots \times I_d \times J_d$ and thus reshape it again to $I_1 J_1 \times I_2 J_2 \times \ldots \times I_d J_d$, which is a $d$th-order tensor. The first mode corresponds to small-scale patches of size $I_1 \times J_1$, while the $d$th-mode corresponds to large-scale partition of whole image as $I_d \times J_d$. Based on this tensorization operations, TR decomposition is able to capture the intrinsic structure information and provides a more compact representation. As shown in Table 4, for 2D matrix case, SVD, TT and TR give exactly same results. In contrast, for 4th-order tensorization cases, TT needs only half number of parameters (2 times compression rate) while TR achieves 3 times compression rate, given the same approximation error 0.1. It should be

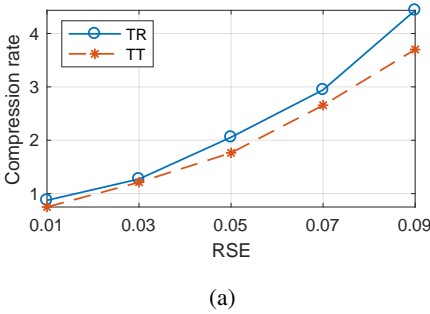
(a)

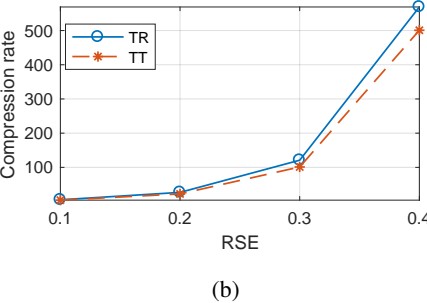
(b)

Figure 5: The comparisons of compression rate and approximation error (RSE) on CIFAR-10 dataset by using TR and TT models.

noted that TR representation provides significantly high compression ability as compared to TT. In addition, Fig. 4 shows TR-SGD results on 'Lena' image by sampling different fraction of data points.

Table 4: Image representation by using tensorization and TR decomposition. The number of parameters is compared for SVD, TT and TR given the same approximation errors.

| Data | $\epsilon = 0.1$ | | $\epsilon = 0.01$ | | $\epsilon = 9e - 4$ | | $\epsilon = 2e - 15$ | |
|---|---|---|---|---|---|---|---|---|
| $n = 256, d = 2$ | SVD | TT/TR | SVD | TT/TR | SVD | TT/TR | SVD | TT/TR |
| | 9.7e3 | 9.7e3 | 7.2e4 | 7.2e4 | 1.2e5 | 1.2e5 | 1.3e5 | 1.3e5 |

| Tensorization | $\epsilon = 0.1$ | | $\epsilon = 0.01$ | | $\epsilon = 2e - 3$ | | $\epsilon = 1e - 14$ | |
|---|---|---|---|---|---|---|---|---|
| | TT | TR | TT | TR | TT | TR | TT | TR |
| $n = 16, d = 4$ | 5.1e3 | 3.8e3 | 6.8e4 | 6.4e4 | 1.0e5 | 7.3e4 | 1.3e5 | 7.4e4 |
| $n = 4, d = 8$ | 4.8e3 | 4.3e3 | 7.8e4 | 7.8e4 | 1.1e5 | 9.8e4 | 1.3e5 | 1.0e5 |
| $n = 2, d = 16$ | 7.4e3 | 7.4e3 | 1.0e5 | 1.0e5 | 1.5e5 | 1.5e5 | 1.7e5 | 1.7e5 |

## 5.3 CIFAR-10

The CIFAR-10 dataset consists of 60000 $32 \times 32$ colour images. We randomly pick up 1000 images for testing of TR decomposition algorithms. As shown in Fig. 5, TR model outperforms TT model in terms of compression rate given the same approximation error, which is caused by strict limitation that the mode-1 rank must be 1 for TT model. In addition, TR is a more generalized model, which contains TT as a special case, thus yielding better low-rank approximation. Moreover, all other TR algorithms can also achieve similar results. The detailed results for $\epsilon = 1e - 1$ are shown in Table 5. Note that TR-SGD can achieve the same performance as TR-ALS, which demonstrates its effectiveness on real-world dataset. Due to high computational efficiency of TR-SGD per epoch, it can be potentially applied to very large-scale dataset. For visualization, TR-SGD results after 10 and 100 epoch are shown in Fig. 6.

## 5.4 TENSORIZING NEURAL NETWORKS USING TR REPRESENTATION

TT representations have been successfully applied to deep neural networks (Novikov et al., 2015), which can significantly reduce the number of model parameters and improve computational efficiency. To investigate the properties of TR representation, we applied TR framework to approximate the weight matrix of a fully-connected layer and compared with TT representation. We run the experiment on the MNIST dataset for the task of handwritten-digit recognition. The same setting of neural network (two fully-connected layers with ReLU activation function) as in (Novikov et al., 2015) was applied for comparisons. The input layer is tensorized to a 4th-order tensor of size $4 \times 8 \times 8 \times 4$, the weight matrix of size $1024 \times 625$ is represented by a TR format of size $4 \times 8 \times 8 \times 4 \times 5 \times 5 \times 5 \times 5$. Through deriving the gradients over each core tensor, all computations can be performed on small core tensors instead of the dense weight matrix by using properties in Sec. 4, yielding the significant improvements of computational efficiency. The experimental results are shown in Fig. 7. We observe that the TR-layer provides much better flexibility than TT-layer, leading to much lower training and

Table 5: Results on CIFAR-10 images.

|          | $\epsilon$ | Ranks | $N_p$ | Epoch |
|----------|-----------|--------------|--------|-------|
| TT-SVD   | 0.092     | (1 7 79 67)  | 66099  | NaN   |
| TR-SVD   | 0.095     | (5,3,49,58)  | 42710  | NaN   |
| TR-BALS  | 0.094     | (61,13,3,6)  | 63278  | 23    |
| TR-ALS   | 0.1076    | (5,3,49,58)  | 42710  | 10    |
| TR-SGD   | 0.1041    | (5,3,49,58)  | 42710  | 100   |

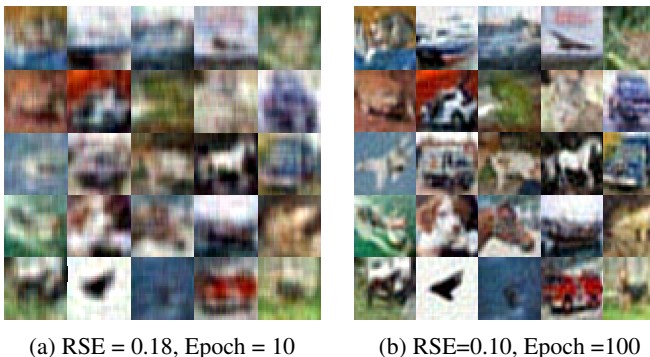

(a) RSE = 0.18, Epoch = 10        (b) RSE=0.10, Epoch =100

Figure 6: The reconstructed images by using TR-SGD after 10 and 100 epochs.

testing errors under the same compression level (i.e., TT/TR ranks). In addition, TR can achieve much better compression rate under the same level of test error. When $r_1 = \ldots = r_4 = 2$, the compression rate of dense weight matrix is up to 1300 times.

We tested the tensorizing neural networks with the same architecture on SVHN dataset (http://ufldl.stanford.edu/housenumbers/). By setting all the TT-ranks in the network to 4, we achieved the test error of 0.13 with compression rate of 444 times, while we can achieve the same test error by setting all the TR-ranks to 3 with compression rate of 592 times. We can conclude that the TR representation can obtain significantly higher compression rate under the same level of test error.

## 6 CONCLUSION

We have proposed a novel tensor decomposition model, which provides an efficient representation for a very high-order tensor by a sequence of low-dimensional cores. The number of parameters in

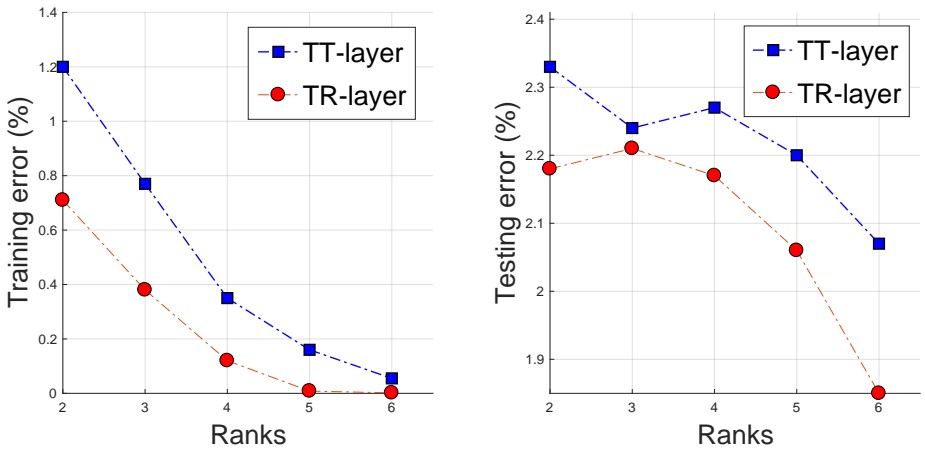

Figure 7: The classification performances of tensorizing neural networks by using TR representation.

our model scales only linearly to the tensor order. To optimize the latent cores, we have presented several different algorithms: TR-SVD is a non-recursive algorithm that is stable and efficient, while TR-BALS can learn a more compact representation with adaptive TR-ranks, TR-SGD is a scalable algorithm which can be also used for tensor completion and online learning. Furthermore, we have investigated the properties on how the basic multilinear algebra can be performed efficiently by operations over TR representations (i.e., cores), which provides a powerful framework for processing large-scale data. The experimental results verified the effectiveness of our proposed algorithms.

ACKNOWLEDGMENTS

This work was partially supported by JSPS KAKENHI (Grant No. 17K00326) and JST CREST (Grant No. JPMJCR1784), Japan and by the Ministry of Education and Science of the Russian Federation (grant 14.756.31.0001).

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

## A  RELATION TO OTHER MODELS

In this section, we discuss the relations between TR model and the classical tensor decompositions including CPD, Tucker and TT models. All these tensor decompositions can be viewed as the transformed representation of a given tensor. The number of parameters in CPD is $\mathcal{O}(dnr)$ that is linear to tensor order, however, its optimization problem is difficult and convergence is slow. The Tucker model is stable and can approximate an arbitrary tensor as close as possible, however, its number of parameters is $\mathcal{O}(dnr + r^d)$ that is exponential to tensor order. In contrast, TT and TR decompositions have similar representation power to Tucker model, while their number of paramters is $\mathcal{O}(dnr^2)$ that is linear to tensor order.

### A.1  CP DECOMPOSITION

The cannonical polyadic decomposition (CPD) aims to represent a $d$th-order tensor $\mathcal{T}$ by a sum of rank-one tensors, given by

$$\mathcal{T} = \sum_{\alpha=1}^{r} \mathbf{u}_{\alpha}^{(1)} \circ \cdots \circ \mathbf{u}_{\alpha}^{(d)}, \tag{13}$$

where each rank-one tensor is represented by an outer product of $d$ vectors. It can be also written in the element-wise form given by

$$T(i_1, \ldots, i_d) = \left\langle \mathbf{u}_{i_1}^{(1)}, \ldots, \mathbf{u}_{i_d}^{(d)} \right\rangle, \tag{14}$$

where $\langle \cdot, \ldots, \cdot \rangle$ denotes an inner product of a set of vectors, i.e., $\mathbf{u}_{i_k}^{(k)} \in \mathbb{R}^r, k = 1, \ldots, d$.

By defining $\mathbf{V}_k(i_k) = \mathrm{diag}(\mathbf{u}_{i_k}^{(k)})$ which is a diagonal matrix for each fixed $i_k$ and $k$, where $k = 1, \ldots, d, i_k = 1, \ldots, n_k$, we can rewrite (14) as

$$T(i_1, \ldots, i_d) = \mathrm{Tr}(\mathbf{V}_1(i_1)\mathbf{V}_2(i_2) \cdots \mathbf{V}_d(i_d)). \tag{15}$$

Hence, CPD can be viewed as a special case of TR decomposition $\mathcal{T} = \Re(\mathcal{V}_1, \ldots, \mathcal{V}_d)$ where the cores $\mathcal{V}_k, k = 1, \ldots, d$ are of size $r \times n_k \times r$ and each lateral slice matrix $\mathbf{V}_k(i_k)$ is a diagonal matrix of size $r \times r$.

### A.2  TUCKER DECOMPOSITION

The Tucker decomposition aims to represent a $d$th-order tensor $\mathcal{T}$ by a multilinear product between a core tensor $\mathcal{G} \in \mathbb{R}^{r_1 \times \cdots \times r_d}$ and factor matrices $\mathbf{U}^{(k)} \in \mathbb{R}^{n_k \times r_k}, k = 1, \ldots, d$, which is expressed by

$$\mathcal{T} = \mathcal{G} \times_1 \mathbf{U}^{(1)} \times_2 \cdots \times_d \mathbf{U}^{(d)} = [\![\mathcal{G}, \mathbf{U}^{(1)}, \ldots, \mathbf{U}^{(d)}]\!]. \tag{16}$$

By assuming the core tensor $\mathcal{G}$ can be represented by a TR decomposition $\mathcal{G} = \Re(\mathcal{V}_1, \ldots, \mathcal{V}_d)$, the Tucker decomposition (16) in the element-wise form can be rewritten as

$$
\begin{aligned}
T(i_1, &\ldots, i_d) \\
&= \Re(\mathcal{V}_1, \ldots, \mathcal{V}_d) \times_1 \mathbf{u}^{(1)T}(i_1) \times_2 \cdots \times_d \mathbf{u}^{(d)T}(i_d) \\
&= \mathrm{Tr}\left\{ \prod_{k=1}^{d} \left( \sum_{\alpha_k=1}^{r_k} \mathbf{V}_k(\alpha_k) u^{(k)}(i_k, \alpha_k) \right) \right\} \\
&= \mathrm{Tr}\left\{ \prod_{k=1}^{d} \left( \mathcal{V}_k \times_2 \mathbf{u}^{(k)T}(i_k) \right) \right\},
\end{aligned}
\tag{17}
$$

where the second step is derived by applying Theorem 2. Hence, Tucker model can be represented as a TR decomposition $\mathcal{T} = \Re(\mathcal{Z}_1, \ldots, \mathcal{Z}_d)$ where the cores are computed by the multilinear products between TR cores representing $\mathcal{G}$ and the factor matrices, respectively, which is

$$\mathcal{Z}_k = \mathcal{V}_k \times_2 \mathbf{U}^{(k)}, \quad k = 1, \ldots, d. \tag{18}$$

## A.3 TT DECOMPOSITION

The tensor train decomposition aims to represent a $d$th-order tensor $\mathcal{T}$ by a sequence of cores $\mathcal{G}_k, k = 1, \ldots, d$, where the first core $\mathbf{G}_1 \in \mathbb{R}^{n_1 \times r_2}$ and the last core $\mathbf{G}_d \in \mathbb{R}^{r_d \times n_d}$ are matrices while the other cores $\mathcal{G}_k \in \mathbb{R}^{r_k \times n_k \times r_{k+1}}, k = 2, \ldots, d - 1$ are 3rd-order tensors. Specifically, TT decomposition in the element-wise form is expressed as

$$T(i_1, \ldots, i_d) = \mathbf{g}_1(i_1)^T \mathbf{G}_2(i_2) \cdots \mathbf{G}_{d-1}(i_{d-1}) \mathbf{g}_d(i_d), \tag{19}$$

where $\mathbf{g}_1(i_1)$ is the $i_1$th row vector of $\mathbf{G}_1$, $\mathbf{g}_d(i_d)$ is the $i_d$th column vector of $\mathbf{G}_d$, and $\mathbf{G}_k(i_k), k = 2, \ldots, d - 1$ are the $i_k$th lateral slice matrices of $\mathcal{G}_k$.

According to the definition of TR decomposition in (1), it is obvious that TT decomposition is a special case of TR decomposition where the first and the last cores are matrices, i.e., $r_1 = r_{d+1} = 1$. On the other hand, TR decomposition can be also rewritten as

$$\begin{aligned} T(i_1, \ldots, i_d) &= \text{Tr}\left\{\mathbf{Z}_1(i_1)\mathbf{Z}_2(i_2) \cdots \mathbf{Z}_d(i_d)\right\} \\ &= \sum_{\alpha_1=1}^{r_1} \mathbf{z}_1(\alpha_1, i_1, :)^T \mathbf{Z}_2(i_2) \cdots \mathbf{Z}_{d-1}(i_{d-1}) \mathbf{z}_d(:, i_d, \alpha_1) \end{aligned} \tag{20}$$

where $\mathbf{z}_1(\alpha_1, i_1, :) \in \mathbb{R}^{r_2}$ is the $\alpha_1$th row vector of the matrix $\mathbf{Z}_1(i_1)$ and $\mathbf{z}_d(:, i_d, \alpha_1)$ is the $\alpha_1$th column vector of the matrix $\mathbf{Z}_d(i_d)$. Therefore, TR decomposition can be interpreted as a sum of TT representations. The number of TT representations is $r_1$ and these TT representations have the common cores $\mathcal{Z}_k$, for $k = 2, \ldots, d - 1$. In general, TR outperforms TT in terms of representation power due to the fact of linear combinations of a group of TT representations. Furthermore, given a specific approximation level, TR representation requires smaller ranks than TT representation.

## B PROOFS

### B.1 PROOF OF THEOREM 1

*Proof.* It is obvious that (1) can be rewritten as

$$\begin{aligned} T(i_1, i_2, \ldots, i_d) &= \text{Tr}(\mathbf{Z}_2(i_2), \mathbf{Z}_3(i_3), \ldots, \mathbf{Z}_d(i_d), \mathbf{Z}_1(i_1)) \\ &= \cdots = \text{Tr}(\mathbf{Z}_d(i_d), \mathbf{Z}_1(i_1), \ldots, \mathbf{Z}_{d-1}(i_{d-1})). \end{aligned}$$

Therefore, we have $\overleftarrow{\mathcal{T}}^k = \Re(\mathcal{Z}_{k+1}, \ldots, \mathcal{Z}_d, \mathcal{Z}_1, \ldots, \mathcal{Z}_k)$. $\qquad\square$

### B.2 PROOF OF PROPERTY 1

*Proof.* According to the definition of TR decomposition, and the cores shown in (10), the $(i_1, \ldots, i_d)$th element of tensor $\mathcal{T}_3$ can be written as

$$T_3(i_1, \ldots, i_d) = \text{Tr}\left( \begin{array}{cc} \prod_{k=1}^d \mathbf{Z}_k(i_k) & 0 \\ 0 & \prod_{k=1}^d \mathbf{Y}_k(i_k) \end{array} \right) = \text{Tr}\left( \prod_{k=1}^d \mathbf{Z}_k(i_k) \right) + \text{Tr}\left( \prod_{k=1}^d \mathbf{Y}_k(i_k) \right).$$

Hence, the *addition* of tensors in the TR format can be performed by merging of their cores. $\qquad\square$

### B.3 PROOF OF PROPERTY 2

*Proof.* The *multilinear product* between a tensor and vectors can be expressed by

$$\begin{aligned} c &= \sum_{i_1, \ldots, i_d} T(i_1, \ldots, i_d) u_1(i_1) \cdots u_d(i_d) = \sum_{i_1, \ldots, i_d} \text{Tr}\left( \prod_{k=1}^d \mathbf{Z}_k(i_k) \right) u_1(i_1) \cdots u_d(i_d) \\ &= \text{Tr}\left( \prod_{k=1}^d \left( \sum_{i_k=1}^{n_k} \mathbf{Z}_k(i_k) u_k(i_k) \right) \right). \end{aligned}$$

Thus, it can be written as a TR decomposition shown in (11) where each core $\mathbf{X}_k \in \mathbb{R}^{r_k \times r_{k+1}}$ becomes a matrix. The computational complexity is equal to $\mathcal{O}(dnr^2)$. $\qquad\square$

---

**Algorithm 1** TR-SVD

---

**Input:** A $d$th-order tensor $\mathcal{T}$ of size $(n_1 \times \cdots \times n_d)$ and the prescribed relative error $\epsilon_p$.
**Output:** Cores $\mathcal{Z}_k, k = 1, \ldots, d$ of TR decomposition and the TR-ranks $\mathbf{r}$.
1: Compute truncation threshold $\delta_k$ for $k = 1$ and $k > 1$.
2: Choose one mode as the start point (e.g., the first mode) and obtain the 1-unfolding matrix $\mathbf{T}_{\langle 1 \rangle}$.
3: Low-rank approximation by applying $\delta_1$-truncated SVD: $\mathbf{T}_{\langle 1 \rangle} = \mathbf{U}\Sigma\mathbf{V}^T + \mathbf{E}_1$.
4: Split ranks $r_1, r_2$ by $\min_{r_1, r_2} \quad \|r_1 - r_2\|$, s. t. $r_1 r_2 = \text{rank}_{\delta_1}(\mathbf{T}_{\langle 1 \rangle})$.
5: $\mathcal{Z}_1 \leftarrow \text{permute}(\text{reshape}(\mathbf{U}, [n_1, r_1, r_2]), [2, 1, 3])$.
6: $\mathcal{Z}^{>1} \leftarrow \text{permute}(\text{reshape}(\Sigma\mathbf{V}^T, [r_1, r_2, \prod_{j=2}^d n_j]), [2, 3, 1])$.
7: **for** $k = 2$ to $d - 1$ **do**
8: $\quad \mathbf{Z}^{>k-1} = \text{reshape}(\mathcal{Z}^{>k-1}, [r_k n_k, n_{k+1} \cdots n_d r_1])$.
9: $\quad$ Compute $\delta_k$-truncated SVD: $\mathbf{Z}^{>k-1} = \mathbf{U}\Sigma\mathbf{V}^T + \mathbf{E}_k$.
10: $\quad r_{k+1} \leftarrow \text{rank}_{\delta_k}(\mathbf{Z}^{>k-1})$.
11: $\quad \mathcal{Z}_k \leftarrow \text{reshape}(\mathbf{U}, [r_k, n_k, r_{k+1}])$.
12: $\quad \mathcal{Z}^{>k} \leftarrow \text{reshape}(\Sigma\mathbf{V}^T, [r_{k+1}, \prod_{j=k+1}^d n_j, r_1])$.
13: **end for**

---

### B.4 PROOF OF PROPERTY 3

*Proof.* Each element in tensor $\mathcal{T}_3$ can be written as

$$
T_3(i_1, \ldots, i_d) = \text{Tr}\left(\prod_{k=1}^d \mathbf{Z}_k(i_k)\right) \text{Tr}\left(\prod_{k=1}^d \mathbf{Y}_k(i_k)\right) = \text{Tr}\left\{\left(\prod_{k=1}^d \mathbf{Z}_k(i_k)\right) \otimes \left(\prod_{k=1}^d \mathbf{Y}_k(i_k)\right)\right\}
$$

$$
= \text{Tr}\left\{\prod_{k=1}^d \left(\mathbf{Z}_k(i_k) \otimes \mathbf{Y}_k(i_k)\right)\right\}.
$$

Hence, $\mathcal{T}_3$ can be also represented as TR format with its cores computed by (12), which costs $\mathcal{O}(dnq^2)$.

Furthermore, one can compute the *inner product* of two tensors in TR representations. For two tensors $\mathcal{T}_1$ and $\mathcal{T}_2$, it is defined as $\langle \mathcal{T}_1, \mathcal{T}_2 \rangle = \sum_{i_1, \ldots, i_d} T_3(i_1, \ldots, i_d)$, where $\mathcal{T}_3 = \mathcal{T}_1 \circledast \mathcal{T}_2$. Thus, the inner product can be computed by applying the Hadamard product and then computing the multilinear product between $\mathcal{T}_3$ and vectors of all ones, i.e., $\mathcal{T}_3 \times_1 \mathbf{u}_1^T \times_2 \cdots \times_d \mathbf{u}_d^T$, where $\mathbf{u}_k = \mathbf{1}, k = 1, \ldots, d$, which can be computed efficiently by using Property 2. $\quad\square$

## C ALGORITHMS

Pseudo codes of TR-SVD and TR-BALS are provided in Alg. 1 and Alg. 2, respectively.

## D ADDITIONAL EXPERIMENTAL RESULTS

### D.1 COIL-100 DATASET

The Columbia Object Image Libraries (COIL)-100 dataset (Nayar et al., 1996) contains 7200 color images of 100 objects (72 images per object) with different reflectance and complex geometric characteristics. Each image can be represented by a 3rd-order tensor of size $128 \times 128 \times 3$ and then is downsampled to $32 \times 32 \times 3$. Hence, the dataset can be finally organized as a 4th-order tensor of size $32 \times 32 \times 3 \times 7200$. The number of features is determined by $r_4 \times r_1$, while the flexibility of subspace bases is determined by $r_2, r_3$. Subsequently, we apply the K-nearest neighbor (KNN) classifier with K=1 for classification. For detailed comparisons, we randomly select a certain ratio $\rho = 50\%$ or $\rho = 10\%$ samples as the training set and the rest as the test set. The classification performance is averaged over 10 times of random splitting. In Table 6, $r_{max}$ of TR decompositions is much smaller than that of TT-SVD. It should be noted that TR representation, as compared to TT, can obtain more compact and discriminant representations. Fig. 8 shows the reconstructed images under different approximation levels.

---

**Algorithm 2** TR-BALS

---

**Input:** A $d$-dimensional tensor $\mathcal{T}$ of size $(n_1 \times \cdots \times n_d)$ and the prescribed relative error $\epsilon_p$.
**Output:** Cores $\mathcal{Z}_k$ and TR-ranks $r_k$, $k = 1, \dots, d$.
  1: Initialize $r_k = 1$ for $k = 1, \dots, d$.
  2: Initialize $\mathcal{Z}_k \in \mathbb{R}^{r_k \times n_k \times r_{k+1}}$ for $k = 1, \dots, d$.
  3: **repeat**   $k \in \text{circular}\{1, 2, \dots, d\}$;
  4:     Compute the subchain $\mathcal{Z}^{\neq(k,k+1)}$.
  5:     Obtain the mode-2 unfolding matrix $\mathbf{Z}_{[2]}^{\neq(k,k+1)}$ of size $\prod_{j=1}^{d} n_j/(n_k n_{k+1}) \times r_k r_{k+2}$.
  6:     $\mathbf{Z}_{(2)}^{(k,k+1)} \leftarrow \arg\min \left\| \mathbf{T}_{[k]} - \mathbf{Z}_{(2)}^{(k,k+1)} \left( \mathbf{Z}_{[2]}^{\neq(k,k+1)} \right)^T \right\|_F$.
  7:     Tensorization of mode-2 unfolding matrix
$$\mathcal{Z}^{(k,k+1)} \leftarrow \text{folding}(\mathbf{Z}_{(2)}^{(k,k+1)}).$$
  8:     Reshape the block core by
$$\tilde{\mathbf{Z}}^{(k,k+1)} \leftarrow \text{reshape}(\mathcal{Z}^{(k,k+1)}, [r_k n_k \times n_{k+1} r_{k+2}]).$$
  9:     Low-rank approximation by $\delta$-truncated SVD $\tilde{\mathbf{Z}}^{(k,k+1)} = \mathbf{U}\boldsymbol{\Sigma}\mathbf{V}^T$.
 10:     $\mathcal{Z}_k \leftarrow \text{reshape}(\mathbf{U}, [r_k, n_k, r_{k+1}])$.
 11:     $\mathcal{Z}_{k+1} \leftarrow \text{reshape}(\boldsymbol{\Sigma}\mathbf{V}^T, [r_{k+1}, n_{k+1}, r_{k+2}])$.
 12:     $r_{k+1} \leftarrow \text{rank}_\delta(\tilde{\mathbf{Z}}^{(k,k+1)})$.
 13:     $k \leftarrow k + 1$.
 14: **until** The desired approximation accuracy is achieved, i.e., $\epsilon \leq \epsilon_p$.

---

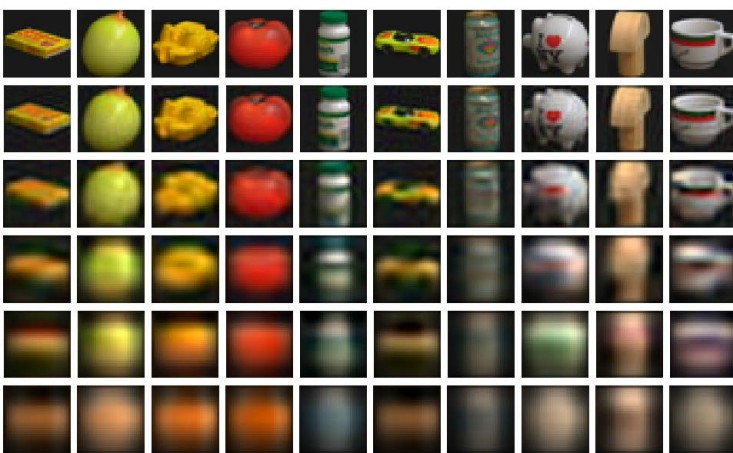

Figure 8: The reconstruction of Coil-100 dataset by using TR-SVD. The top row shows the original images, while the reconstructed images are shown from the second to sixth rows corresponding to $\epsilon$=0.1, 0.2, 0.3, 0.4, 0.5, respectively.

## D.2  KTH VIDEO DATASET

We test the TR representation for KTH video database (Laptev & Lindeberg, 2006) containing six types of human actions (walking, jogging, running, boxing, hand waving and hand clapping) performed several times by 25 subjects in four different scenarios: outdoors, outdoors with scale variation, outdoors with different clothes and indoors as illustrated in Fig. 9. There are 600 video sequences for each combination of 25 subjects, 6 actions and 4 scenarios. Each video sequence was downsampled to $20 \times 20 \times 32$. Finally, we can organize the dataset as a tensor of size $20 \times 20 \times 32 \times 600$. For extensive comparisons, we choose different error bound $\epsilon_p \in \{0.2, 0.3, 0.4\}$. In Table 7, we can see that TR representations achieve better compression ratio reflected by smaller $r_{max}, \bar{r}$ than that of TT-SVD, while TT-SVD achieves better compression ratio than CP-ALS. For instance, when

Table 6: The comparisons of different algorithms on Coil-100 dataset. $\epsilon$, $r_{max}$,$\bar{r}$ denote relative error, the maximum rank and the average rank, respectively.

| | $\epsilon$ | $r_{max}$ | $\bar{r}$ | Acc. (%) ($\rho = 50\%$) | Acc. (%) ($\rho = 10\%$) |
|---|---|---|---|---|---|
| | 0.19 | 67 | 47.3 | 99.05 | 89.11 |
| | 0.28 | 23 | 16.3 | 98.99 | 88.45 |
| TT-SVD | 0.37 | 8 | 6.3 | 96.29 | 86.02 |
| | 0.46 | 3 | 2.7 | 47.78 | 44.00 |
| | 0.19 | 23 | 12.0 | 99.14 | 89.29 |
| | 0.28 | 10 | 6.0 | 99.19 | 89.89 |
| TR-SVD | 0.36 | 5 | 3.5 | 98.51 | 88.10 |
| | 0.43 | 3 | 2.3 | 83.43 | 73.20 |

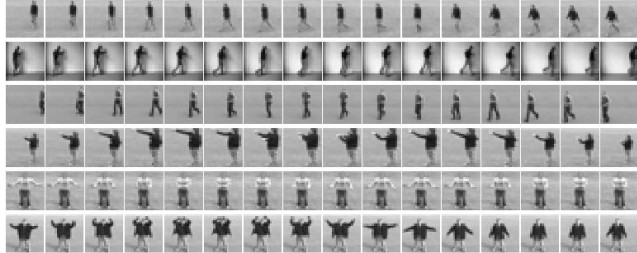

Figure 9: Video dataset consists of six types of human actions performed by 25 subjects in four different scenarios. From the top to bottom, six video examples corresponding to each type of actions are shown.

$\epsilon \approx 0.2$, CP-ALS requires $r_{max} = 300$, $\bar{r} = 300$; TT-SVD requires $r_{max} = 139$, $\bar{r} = 78$, while TR-SVD only requires $r_{max} = 99$, $\bar{r} = 34.2$. For classification performance, we observe that the best accuracy ($5 \times 5$-fold cross validation) achieved by CP-ALS, TT-SVD, TR-SVD are 80.8%, 84.8%, 87.7%, respectively. Note that these classification performances might not be the state-of-the-art on this dataset, we mainly focus on the comparisons of representation ability among CP, TT, and TR decomposition frameworks. To obtain the best performance, we may apply the powerful feature extraction methods to TT or TR representations of dataset. It should be noted that TR decompositions achieve the best classification accuracy when $\epsilon = 0.29$, while TT-SVD and CP-ALS achieve their best classification accuracy when $\epsilon = 0.2$. This indicates that TR decomposition can preserve more discriminant information even when the approximation error is relatively high. This experiment demonstrates that TR decompositions are effective for unsupervised feature representation due to their flexibility of TR-ranks and high compression ability.

Table 7: The comparisons of different algorithms on KTH dataset. $\epsilon$ denotes the obtained relative error; $r_{max}$ denotes maximum rank; $\bar{r}$ denotes the average rank; and Acc. is the classification accuracy.

| | $\epsilon$ | $r_{max}$ | $\bar{r}$ | Acc. ($5 \times 5$-fold) |
|---|---|---|---|---|
| | 0.20 | 300 | 300 | 80.8 % |
| CP-ALS | 0.30 | 40 | 40 | 79.3 % |
| | 0.40 | 10 | 10 | 66.8 % |
| | 0.20 | 139 | 78.0 | 84.8 % |
| TT-SVD | 0.29 | 38 | 27.3 | 83.5 % |
| | 0.38 | 14 | 9.3 | 67.8 % |
| | 0.20 | 99 | 34.2 | 78.8 % |
| TR-SVD | 0.29 | 27 | 12.0 | 87.7 % |
| | 0.37 | 10 | 5.8 | 72.4 % |

