# OpenReview forum: "Learning Efficient Tensor Representations with Ring Structure Networks"
_ICLR.cc/2018/Conference — Invite to Workshop Track_

### Official Review · AnonReviewer3 · 2017-11-29
**Preliminary examination. Lacks novelty.**

**Rating:** 5
**Confidence:** 4

**Review:**

This paper proposes a tensor train decomposition with a ring structure for function approximation and data compression. Most of the techniques used are well-known in the tensor community (outside of machine learning). The main contribution of the paper is the introduce such techniques to the ML community and presents experimental results for support.

The paper is rather preliminary in its examination. For example, it is claimed that the proposed decomposition provides "enhanced representation ability", but this is not justified rigorously either via more comprehensive experimentation or via a theoretical justification. Furthermore, the paper lacks in novelty aspect, as it is uses mostly well-known techniques.

---

> ### Author Response · Authors · 2017-12-25
> **Responses and Revisions**
>
> We would like to thank the reviewer for the fruitful comments.
>
> Tensor ring (TR) decomposition is a newly proposed tensor model in the tensor community. Although TR at a glance seems to simply have one more connection to a tensor train (TT) in terms of geometric
> structure, its computation principle and mathematical properties are essentially different. As compared to all well-known tensor models, e.g., CP, Tucker, TT and HT, the tensor ring model is the only one
> which has a loop connection of latent components.
>
> The TR model has several significant advantages over TTs. i) The TR-ranks are much smaller than TT-ranks (ref. Theorem 2), given the same approximation error. ii) For TTs, any permutation of tensor dimensions will yield inconsistent results, and thus the performance of TT models is sensitive to the order of dimensions, while TR models have circular permutation invariance. iii) Due to $r_1=r_d =1$ in TTs, the rank of middle core (r_k) usually need to be very large, while the ranks of TR cores in principle can be equally distributed.
>
> For algorithms, since there is a loop connection in TRs, the developments of TR-SVD and BALS are not trivial as compared to TT-SVD. In particular, BALS enables us to automatically determine the rank between the first and last cores, which is not possible in the TT model. We would like to emphasize that our newly developed SGD algorithm is scalable and efficient by randomly sampling one tensor entry per update, while such an algorithm is not studied for the TT model. Note that TR-SGD can achieve similar results to ALS algorithms by using only 1% tensor entries with each entry used only once (see Table 3 in our revision).
>
> We proved that the convenient computation principles of the TT-format are mostly retained for the TR-format with slightly different operations on cores (see Sec. 4), which adds further novelty.
>
> In experiments, the most impressive results are the compactness of the TR expression. TT can approximate one 2D image (matrix) with $\epsilon= 0.1$ by using 0.53 times parameters of SVD, while TR only needs 0.39 times parameters (see Table 4 in revision).  Even when $\epsilon=1e-15$, TR still needs 0.56 times parameters of SVD while TT needs the equivalent number to SVD. These results are achieved by our proposed tensorization strategy, which is the first time to be studied in the tensor community.
>
> We firstly studied that the physical meaning of TR-cores corresponds to different scales of images by using our tensorization method, which is shown by adding noise to cores (see Fig. 1).
>
> For "enhanced representation ability", we proved that TT can be considered as a special case of TR when we set $r_1=r_d=1$ (see A.3 in Appendix). Thus, TR is more generalized and flexible than TT. Secondly, TR-ranks are much smaller than TT-ranks (from Theorem 2). Thirdly, TR is proved to be a sum of $r_1$ TT formats with common cores $G_k, 2<=k<=d-1$ (see A.3 in Appendix).
>
> The experiment results (Table 4 in revision) showed that TR needs only 0.56 times parameters of TT when $\epsilon=1e-14$. TT always needs more parameters than TR for all settings.  For the CIFAR dataset, Table 5 shows that TT needs 1.5 times parameters of TR.  The results on the Coil-100 and KTH video datasets also showed similar phenomena (see Tables 6 and 7 in Appendix).
>
> In our revision, we added many additional experiments. In Sec. 5.4, we applied TR representation to approximate the dense weight matrix of fully-connected layer in neural networks. By deriving the learning algorithm over each small core, all computations can be performed by using core tensors instead of full tensor, yielding improved computation efficiency. In addition, we compared the performance of TT-layer and TR-layer in Fig. 7. The tensorizing neural networks are tested on MNIST and SVHN datasets.  In Sec. 5.3, we added extensive experimental comparisons (compression rate vs. approximation error) as shown in Fig. 5. In Sec. 5.1, we added more experiments for investigating the benefit of circular shift invariance of TR, as shown in Table 2.

---

### Official Review · AnonReviewer2 · 2017-11-30

**Rating:** 6
**Confidence:** 3

**Review:**

This paper presents a tensor decomposition method called tensor ring (TR) decomposition. The proposed decomposition approximates each tensor element via a trace operation over the sequential multilinear products of lower order core tensors. This is in contrast with another popular approach based on tensor train (TT) decomposition which requires several constraints on the core tensors (such as the rank of the first and last core tensor to be 1).

To learn TR representations, the paper presents a non-iterative TR-SVD algorithm that is similar to TT-SVD algorithm. To find the optimal lower TR-ranks, a block-wise ALS algorithms is presented, and an SGD algorithm is also presented to make the model scalable.

The proposed method is compared against the TT method on some synthetic high order tensors and on an image completion task, and shown to yield better results.

This is an interesting work. TT decompositions have gained popularity in the tensor factorization literature recently and the paper tries to address some of their key limitations. This seems to be a good direction. The experimental results are somewhat limited but the overall framework looks appealing.

---

> ### Author Response · Authors · 2017-12-25
> **Thank you for positive evaluations.**
>
> Thank you very much for your positive evaluation.
>
> In our revision, we added many additional experiments. In Sec. 5.4, we applied TR representation to approximate the dense weight matrix of fully-connected layer in neural networks. By deriving the learning algorithm over each small core, all computations can be performed by using core tensors instead of full tensor, yielding improved computation efficiency. In addition, we compared the performance of TT-layer and TR-layer in Fig. 7. The tensorizing neural networks are tested on MNIST and SVHN datasets.  In Sec. 5.3, we added extensive experimental comparisons (compression rate vs. approximation error) as shown in Fig. 5. In Sec. 5.1, we added more experiments for investigating the benefit of circular shift invariance of TR, as shown in Table 2.

---

### Official Review · AnonReviewer5 · 2017-12-17
**Interesting but limited contribution and validation**

**Rating:** 5
**Confidence:** 4

**Review:**

The paper addresses the problem of tensor decomposition which is relevant and interesting. The paper proposes Tensor Ring (TR) decomposition which improves over and bases on the Tensor Train (TT) decomposition method. TT decomposes a tensor in to a sequences of latent tensors where the first and last tensors are a 2D matrices.

The proposed TR method generalizes TT in that the first and last tensors are also 3rd-order tensors instead of 2nd-order. I think such generalization is interesting but the innovation seems to be very limited.

The paper develops three different kinds of solvers for TR decomposition, i.e., SVD, ALS and SGD. All of these are well known methods.

Finally, the paper provides experimental results on synthetic data (3 oscillated functions) and image data (few sampled images). I think the paper could be greatly improved by providing more experiments and ablations to validate the benefits of the proposed methods.

Please refer to below for more comments and questions.

-- The rating has been updated.

Pros:
1. The topic is interesting.
2. The generalization over TT makes sense.

Cons:
1. The writing of the paper could be improved and more clear: the conclusions on inner product and F-norm can be integrated into "Theorem 5". And those "theorems" in section 4 are just some properties from previous definitions; they are not theorems.
2. The property of TR decomposition is that the tensors can be shifted (circular invariance). This is an interesting property and it seems to be the major strength of TR over TT. I think the paper could be significantly improved by providing more applications of this property in both theory and experiments.
3. As the number of latent tensors increase, the ALS method becomes much worse approximation of the original optimization. Any insights or results on the optimization performance vs. the number of latent tensors?
4. Also, the paper mentions Eq. 5 (ALS) is optimized by solving d subproblems alternatively. I think this only contains a single round of optimization. Should ALS be applied repeated (each round solves d problems) until convergence?
5. What is the memory consumption for different solvers?
6. SGD also needs to update at least d times for all d latent tensors. Why is the complexity O(r^3) independent of the parameter d?
7. The ALS is so slow (if looking at the results in section 5.1), which becomes not practical. The experimental part could be improved by providing more results and description about a guidance on how to choose from different solvers.
8. What does "iteration" mean in experimental results such as table 2? Different algorithms have different cost for "each iteration" so comparing that seems not fair. The results could make more sense by providing total time consumptions and time cost per iteration. also applies to table 4.
9. Why is the \epsion in table 3 not consistent? Why not choose \epsion = 9e-4 and \epsilon=2e-15 for tensorization?
10. Also, table 3 could be greatly improved by providing more ablations such as results for (n=16, d=8), (n=4, d=4), etc. That could help readers to better understand the effect of TR.
11. Section 5.3 could be improved by providing a curve (compression vs. error) instead of just providing a table of sampled operating points.
12. The paper mentions the application of image representation but only experiment on 32x32 images. How does the proposed method handle large images? Otherwise, it does not seem to be a practical application.
13. Figure 5: Are the RSE measures computed over the whole CIFAR-10 dataset or the displayed images?

Minor:
- Typo: Page 4 Line 7 "Note that this algorithm use the similar strategy": use -> uses

---

> ### Author Response · Authors · 2017-12-25
> **Responses and Revisions**
>
> We would like to thank the reviewer for the constructive and insightful comments.
>
> The proposed TR generalizes TT not only in that the first and last tensors are 3rd-order instead of 2nd-order, but also a tensor contraction (link) between the first tensor and last tensor is added. This additional operation makes TR having essentially different computation principle and mathematical properties. As a result, TR has several advantages over TT such as enhanced representation ability, smaller ranks than TT, circular permutation invariance.
>
> Since there is a loop connection in TR, the developments of TR-SVD and TR-BALS are not trivial as compared to TT. In particular, BALS enables us to automatically determine the rank between the first and last cores, which is not possible in the TT model.  Although ALS and SGD are well-known methods, our novelty is how we can apply these standard techniques to solve TR decomposition.
>
> The most notable experiment is to represent an image by using our proposed tensorization and tensor decomposition (sec. 5.2). By converting an image to 4th, 8th, and 16th order tensors, TT/TR can represent it by using much less parameters than gold standard SVD on the original matrix. In addition, TR needs 0.56 times parameters of TT (see Table 4 in revision). This is the first time to show significant advantages of artificially converting a 2D matrix (real image) to a high-order tensor.
>
> As suggested by AnonReviewer5, we added a new experiment on two datasets in our revision (Sec. 5.4). TTs/TRs are applied to approximate a dense weight matrix of fully connected layers in neural networks. The results show that TR can always achieve better training and testing error than TT by using different ranks. TR-layer can achieve much better compression rate than TT-layer under the same level of test error. In particular, the compression factor of parameters by using TR is up to 1300 times.
>
> Response to Cons:
> 1.	We improved the clarity of the paper as AnonReviewer5 suggested.
> 2.	In the revised manuscript, we added an experiment to demonstrate the benefit of this property (see Sec. 5.1 and Table 2).
> 3.	 In Table 3, as the number of latent tensors increases, the original tensor data has totally different size, leading to a slow convergence rate of ALS. In addition, ALS is prone to get stuck in a local minimum. Thus, for large-scale tensors, TR-SGD would be more promising. Here, we mainly show TR-SGD can achieve similar approximation to TR-ALS by using partially observed tensor data.
> 4.	ALS should be applied repeatedly till convergence. We mentioned this in our revision.
> 5.	We provide memory costs for different solvers in our revision (Sec. 3.3).
> 6.	The complexity of TR-SGD should be O(d^2r^3) for each data point.
> 7.	In Sec. 5.1, TT-ALS is so slow due to many iterations for convergence, while TT-SVD is a non-iterative method.  We provided discussions and descriptions about how to choose an appropriate solver in our revision (Sec. 3.3).
> 8.	“Iteration” in Tables 3 and 5 should be changed to “Epoch”, which indicates how many times the whole data tensor is used for optimization. The main point is that TR-SGD can achieve similar error by using 1% of tensor elements. For each epoch, the cost of TR-ALS is O(Ndr^4+dr^6), while the cost of TR-SGD is O(Nd^2r^3).
> 9.	In Table 3, \epsilon is the real achieved error, we choose \epsilon = 1e-1, 1e-2, 1e-3, 1e-15 for all methods, but it is impossible to obtain exactly the same error due to discrete value of ranks.
> 10.	 In Table 4, all results are obtained by using the same image data (256x256), and thus (n=16, d=8) and (n=4, d=4) are impossible, because 16^8 and 4^4 are not equal to 256^2.
> 11.	 We added more experiments shown in Figure 5 in our revision as suggested by AnonReviewer5.
> 12.	 The size of each sample is not a crucial factor, because we consider the whole dataset as one tensor.   We also performed experiments on a larger image (256 x 256) in Sec. 5.2. The tensor in Sec. 5.3 is of size 1000 x 32 x 32 (1million entries). Although each image is small, a tensor of the whole dataset is large. In Appendix, the COIL-100 dataset is a tensor of size 32 x 32 x 3 x 7200 (22 million entries) and the KTH dataset is of size 20 x 20 x 32 x 600 (7 million entries). These tensors are large enough for practical applications. For example, we may use TR to represent a weight matrix of size 640,000 in neural networks (newly added Sec. 5.4).
> 13.	In Figure 6, RSE is computed over the whole CIFAR-10 dataset rather than the displayed images.
>
> Thank you for pointing out the typo in Page 4 Line 7.

---

> > ### Comment · AnonReviewer5 · 2018-01-15
> > **Updated review**
> >
> > Thanks for your thorough rebuttal. I think the quality of the current version is greatly improved. However, considering the technical contribution and the very limited improvement over TT by the proposed method (shown in the new results). I still have a negative feeling about this paper. The rating is upgraded but still in the negative direction.

---

> > > ### Author Response · Authors · 2018-01-28
> > > **Re: Updated review**
> > >
> > > We appreciate the reviewer’s positive feedbacks on our revised paper.
> > >
> > > For experiments on one image,  the compression rate achieved by our method is almost 2 times over TT method.  For tensorizing neural networks, although the classification performance of our method is only slightly better than TT,  given the same testing error 2.2%, the compression rate of our method is 1300, while TT is 290.  As shown in [Novikov, NIPS 2015], the main advantage of TT based tensorizing neural networks is high compressive ability rather than classification performance.  Hence, the improvement over TT is significant.
> > >
> > > Although we employed well-known ALS and SGD techniques, our paper provides new algorithms for tensor ring decomposition.  This paper also provides firstly the detailed theoretical analysis of rank of tensor ring and the mathematic computations of tensor ring format, as well as relationship with other popular tensor models.  These results are very helpful and important for future applications of tensor ring representation to machine learning problems.

---

### Decision · Program_Chairs · 2018-01-29
**ICLR 2018 Conference Acceptance Decision**

**Decision:**

Invite to Workshop Track

**Comment:**

This paper proposes a new way of learning tensors representation with ring-structured decompositions rather than through Tensor Train methods. The paper investigates the mathematical properties of this decomposition and provides synthetic experiments. There was some debate, with the reviewers, about the novelty and impact of this method, where overall the feeling was this work was too preliminary to be accepted. The idea, from my understanding, is interesting and would benefit from discussion at the workshop track, but the authors are investigated to make a stronger case for the novelty of this method in any further work and, in particular, to consider showing empirical improvement on "real" data where TT methods are currently applied.